

# The synergistic effect of concatenation in phylogenomics: the case in *Pantoea*

Marike Palmer[1], Stephanus N. Venter[1], Alistair R. McTaggart[1,2], Martin P.A. Coetzee[1], Stephanie Van Wyk[1], Juanita R. Avontuur[1], Chrizelle W. Beukes[1], Gerda Fourie[1], Quentin C. Santana[1], Magriet A. Van Der Nest[1], Jochen Blom[3] and Emma T. Steenkamp[1]

[1] Department of Biochemistry, Genetics and Microbiology, DST-NRF Centre of Excellence in Tree Health Biotechnology (CTHB) and Forestry and Agricultural Biotechnology Institute (FABI), University of Pretoria, Pretoria, Gauteng, South Africa
[2] Queensland Alliance for Agriculture and Food Innovation, University of Queensland, Brisbane, Queensland, Australia
[3] Bioinformatics and Systems Biology, Justus Liebig Universität Gießen, Giessen, Germany

Corresponding author
Stephanus N. Venter,
fanus.venter@up.ac.za

## ABSTRACT

With the increased availability of genome sequences for bacteria, it has become routine practice to construct genome-based phylogenies. These phylogenies have formed the basis for various taxonomic decisions, especially for resolving problematic relationships between taxa. Despite the popularity of concatenating shared genes to obtain well-supported phylogenies, various issues regarding this combined-evidence approach have been raised. These include the introduction of phylogenetic error into datasets, as well as incongruence due to organism-level evolutionary processes, particularly horizontal gene transfer and incomplete lineage sorting. Because of the huge effect that this could have on phylogenies, we evaluated the impact of phylogenetic conflict caused by organism-level evolutionary processes on the established species phylogeny for *Pantoea*, a member of the *Enterobacterales*. We explored the presence and distribution of phylogenetic conflict at the gene partition and nucleotide levels, by identifying putative inter-lineage recombination events that might have contributed to such conflict. Furthermore, we determined whether smaller, randomly constructed datasets had sufficient signal to reconstruct the current species tree hypothesis or if they would be overshadowed by phylogenetic incongruence. We found that no individual gene tree was fully congruent with the species phylogeny of *Pantoea*, although many of the expected nodes were supported by various individual genes across the genome. Evidence of recombination was found across all lineages within *Pantoea*, and provides support for organism-level evolutionary processes as a potential source of phylogenetic conflict. The phylogenetic signal from at least 70 random genes recovered robust, well-supported phylogenies for the backbone and most species relationships of *Pantoea*, and was unaffected by phylogenetic conflict within the dataset. Furthermore, despite providing limited resolution among taxa at the level of single gene trees, concatenated analyses of genes that were identified as having no signal resulted in a phylogeny that resembled the species phylogeny of *Pantoea*. This distribution of signal and noise across the genome presents the ideal situation for phylogenetic inference, as the topology from a $\geq$70-gene concatenated species phylogeny is not driven by single genes, and our data suggests that this finding may also hold true for smaller datasets. We thus argue that, by using a

concatenation-based approach in phylogenomics, one can obtain robust phylogenies due to the synergistic effect of the combined signal obtained from multiple genes.

# INTRODUCTION

Whole genome sequences are now routinely used for phylogenetic inference, particularly in bacteria (*Abdul Rahman et al., 2016*; *Beukes et al., 2017*; *Callister et al., 2008*; *Gupta, Naushad & Baker, 2015*; *Meehan & Beiko, 2014*; *Palmer et al., 2017*; *Schwartz et al., 2015*; *Zhang et al., 2011*). Many approaches for investigating evolutionary relationships across different taxonomic ranks have been developed (*Daubin, Gouy & Perrière, 2001*; *Daubin, Gouy & Perriere, 2002*; *Jolley et al., 2012*; *Yokono, Satoh & Tanaka, 2018*). These range from alignment-free approaches (*Yokono, Satoh & Tanaka, 2018*) to alignment-based analyses of a small number of highly conserved genes across large numbers of taxa (e.g., different bacterial phyla or orders (*Abdul Rahman et al., 2016*; *Gupta, Naushad & Baker, 2015*; *Jolley et al., 2012*)), to using hundreds or thousands of genes, obtained from whole genome sequences and shared by all members of smaller groups (e.g., species, genus or family (*Meehan & Beiko, 2014*; *Palmer et al., 2017*; *Schwartz et al., 2015*; *Zhang et al., 2011*)).

The use of large numbers of shared genes for phylogenetic inference, referred to here as phylogenomics (*Daubin, Gouy & Perrière, 2001*; *Daubin, Gouy & Perriere, 2002*; *Eisen & Fraser, 2003*; *Kumar et al., 2011*), have been argued to be the most reliable option for recovering a species topology reflective of vertical descent (*Andam & Gogarten, 2011*; *Coenye et al., 2005*; *Daubin, Gouy & Perriere, 2002*; *Galtier & Daubin, 2008*). This is because the massive numbers of characters sampled is thought to dilute phylogenetic conflict within the dataset, to levels where a single robust evolutionary hypothesis is obtainable (*Andam & Gogarten, 2011*; *Coenye et al., 2005*; *Cohan, 2001*; *Daubin, Gouy & Perriere, 2002*; *Galtier & Daubin, 2008*; *Klenk & Göker, 2010*). It has been suggested, particularly in bacteria, that an overall genomic core (the set of genes shared by all members of a group) exists between closely related taxa that remains evolutionarily cohesive (*Coenye et al., 2005*; *Daubin, Gouy & Perriere, 2002*). The signal found within these core genes would thus be the signal for inheritance and would be appropriate for inferring the ancestral relationships (*Daubin, Gouy & Perriere, 2002*).

Despite some evidence for a genomic core (*Callister et al., 2008*; *Daubin, Gouy & Perriere, 2002*; *Grote et al., 2012*; *Sarkar & Guttman, 2004*), numerous studies have shown that the evolutionary trajectory of genes within this subgenomic compartment may be incongruent (*Bapteste et al., 2009*; *Dagan & Martin, 2006*; *Jeffroy et al., 2006*; *Rokas et al., 2003*; *Thiergart, Landan & Martin, 2014*). *Dagan & Martin (2006)* captured this conflict in their "tree of one percent" concept. They referred to research by Ciccarelli and colleagues (*2006*), who used 31 protein sequences to recover a robust phylogenetic hypothesis across a diverse set of bacterial taxa. This was after the removal of sequences harbouring
phylogenetic conflict from a conservative average bacterial genome of 3,000 genes. In other words, the resulting phylogenetic tree that was interpreted as the evolutionary history of the taxa, was based on roughly 1% of the average genome of these taxa (*Dagan & Martin, 2006*). Additionally, research has shown that species trees may in some cases be driven by only a handful of genes, particularly where contradictory species relationships are routinely observed from single gene trees (*Salichos & Rokas, 2013*; *Shen, Hittinger & Rokas, 2017*; *Thiergart, Landan & Martin, 2014*). It is thus still unclear whether employing genome data in a concatenation-based approach is truly an appropriate way of inferring evolutionary relationships, despite the popularity of this approach.

The incongruence often observed between gene and species trees can be attributed to two main factors: phylogenetic errors and organism-level evolutionary processes (*Doyle, 1992*; *Wendel & Doyle, 1998*). Phylogenetic errors (i.e., stochastic errors due to the use of too little information and systematic errors caused by non-phylogenetic signal) during tree inferences are mainly overcome by increased character and taxon sampling (*Hedtke, Townsend & Hillis, 2006*; *Jeffroy et al., 2006*; *Palmer et al., 2017*; *Philippe et al., 2011*; *Pollock et al., 2002*; *Yokono, Satoh & Tanaka, 2018*). Organism-level evolutionary processes can be difficult to account for if they result in different evolutionary histories for genes that cannot be integrated into a single bifurcating evolutionary hypothesis (*Wendel & Doyle, 1998*). When phylogenetic error is excluded, incomplete lineage sorting (ILS) and horizontal gene transfer (HGT) are frequently the primary organism-level processes responsible for phylogenetic incongruence (*Galtier & Daubin, 2008*; *Mallet, Besansky & Hahn, 2016*; *Retchless & Lawrence, 2010*). An ongoing debate in the scientific community is whether to concatenate and risk a well-supported but incorrect species tree that also captures all phylogenetic conflict in a dataset, or pool the phylogenetic signal from hundreds of gene trees in supertree or reconciliation approaches (*Daubin, Gouy & Perrière, 2001*; *Galtier & Daubin, 2008*; *Ren, Tanaka & Yang, 2009*; *Retchless & Lawrence, 2010*; *Sanderson & Driskell, 2003*; *Szöllősi et al., 2012*; *Williams et al., 2017*). Reconciliation approaches efficiently account for HGT because genome evolution is modelled and the data produced are used for quantifying gene transfer and for inferring species trees that accommodate this process (*Szöllősi et al., 2012*).

For this study, the bacterial genus *Pantoea* was used as a model to explore the impact of potentially conflicting signal caused by organism-level evolutionary processes on the current phylogenetic hypothesis for the group. This phylogeny was constructed previously using a concatenation-based approach that accounted for the majority of known phylogenetic errors through Maximum Likelihood analyses of partitioned datasets with appropriate evolutionary models (*Palmer et al., 2017*). *Pantoea* forms part of the family *Erwiniaceae* in the order *Enterobacterales* (*Adeolu et al., 2016*) and is closely related to the genera *Erwinia* and *Tatumella* (*Adeolu et al., 2016*; *Brady et al., 2010b*; *Glaeser & Kämpfer, 2015*; *Palmer et al., 2017*). This genus has been extensively studied and represents a diverse assemblage of organisms that employs an array of different and important lifestyles (*Brady et al., 2010a*; *Brady et al., 2009*; *Brady et al., 2010b*; *Lim et al., 2014*; *Ma et al., 2016*; *Palmer et al., 2016*; *Palmer et al., 2017*; *Walterson & Stavrinides, 2015*). Our three main objectives were to (i) determine whether or not the dataset used to infer hypotheses (based on concatenation

and a multi-species coalescent approach) included phylogenetic conflict, and if so, how this conflict is distributed across the genome; (ii) to determine whether the observed conflicts could be ascribed to organism-level evolutionary processes, such as HGT and ILS; and (iii) to determine whether limited sets of genes contain enough phylogenetic signal to overshadow potential conflict within the dataset in order to obtain phylogenies resembling the species phylogenetic hypotheses for *Pantoea*. To achieve these objectives we investigated conflict at the level of gene partitions and at specific nucleotide sites to detect recombination between the different lineages of the *Pantoea* species phylogeny and also to compare regions that differed significantly in their nucleotide composition to the rest of the alignments.

## MATERIALS AND METHODS

### Dataset preparation

Shared genes for the 27 taxa of interest (Table 1) were determined with the Efficient Database framework for comparative Genome Analyses using BLAST score Ratios (EDGAR) server (*Blom et al., 2016*). The nucleotide sequences for all shared genes were downloaded from the EDGAR server. Subsequently, the combined file of all sequences were split into individual gene files. Multiple sequence alignments of genes were generated with MUSCLE (*Edgar, 2004*) as part of CLC Main Workbench v 7.6 (CLC Bio, Aarhus, Denmark). This was followed by manual inspection and correction of alignments in BioEdit v. 7.0.9 (*Hall, 2011*) to ensure that the correct reading-frame was selected for all genes. Genes were then trimmed in BioEdit to eliminate gene length variation due to potential differences in gene prediction across the different genomes. To generate concatenated datasets, the respective nucleotide and protein sequences were combined with FASconCAT-G v. 1.02 (*Kück & Longo, 2014*).

### Phylogenetic analyses

Approximate maximum likelihood (AML; *Price, Dehal & Arkin, 2010*) analyses were performed on all individual protein sequences, as well as, on the concatenated protein and nucleotide sequence data matrices. For individual gene trees, analyses were performed in a sequential manner, utilising an in-house python script (File S1). For computational efficiency, AML analyses were employed in this study instead of traditional maximum likelihood (ML) analyses in alternate software. Time estimates for the construction of a single gene tree based on ML is ca. 27 minutes/gene (RAxML v. 8.0.20 (*Stamatakis, 2014*)) versus ca. 4 minutes/gene for AML (FastTree v. 2.1), which is not surprising as up to 100 times speed increases were reported previously (*Price, Dehal & Arkin, 2010*). All AML phylogenies were constructed with FastTree v. 2.1 (*Price, Dehal & Arkin, 2010*) using default settings. When the relationships obtained from concatenated AML analyses were not robustly supported (SH-support >0.95), these relationships were verified using RAxML v. 8.0.20 (*Stamatakis, 2014*).

A multi-species coalescent (MSC) approach (*Mirarab et al., 2014*) was employed to construct a species tree from the individual gene phylogenies. This summary method was used to reconstruct a species tree, in the presence of potential ILS (*Mirarab et al., 2014*), by
**Table 1  Genome sequences utilised in this study.**

| Genus | Species | Strain[a] | Accession number[b] | Reference |
|---|---|---|---|---|
| Pantoea | Pantoea agglomerans | R 190 | JNGC00000000.1 | Lim et al. (2014) |
| | Pantoea allii | LMG 24248[T] | MLFE00000000.1 | Palmer et al. (2017) |
| | Pantoea ananatis | LMG 2665[T] | JMJJ00000000.1 | De Maayer et al. (2014) |
| | Pantoea anthophila | 11-2 | JXXL00000000.1 | Wan et al. (2015) |
| | Pantoea brenneri | LMG 5343[T] | MIEI00000000.1 | Palmer et al. (2017) |
| | Pantoea conspicua | LMG 24534[T] | MLFN00000000.1 | Palmer et al. (2017) |
| | Pantoea cypripedii | LMG 2657[T] | MLJI00000000.1 | Palmer et al. (2017) |
| | Pantoea deleyi | LMG 24200[T] | MIPO00000000.1 | Palmer et al. (2017) |
| | Pantoea dispersa | EGD-AAK13 | AVSS00000000.1 | – |
| | Pantoea eucalypti | aB | AEDL00000000.1 | – |
| | Pantoea eucrina | LMG 2781[T] | MIPP00000000.1 | Palmer et al. (2017) |
| | Pantoea rodasii | LMG 26273[T] | MLFP00000000.1 | Palmer et al. (2017) |
| | Pantoea rwandensis | LMG 26275[T] | MLFR00000000.1 | Palmer et al. (2017) |
| | Pantoea septica | LMG 5345[T] | MLJJ00000000.1 | Palmer et al. (2017) |
| | Pantoea stewartii subsp. stewartii | DC 283 | AHIE00000000.1 | – |
| | Pantoea stewartii subsp. indologenes | LMG 2632[T] | JPKO00000000.1 | – |
| | Pantoea vagans | C9-1 | CP001894.1, CP001893.1, CP001894.1 | Smits et al. (2010) |
| | Pantoea wallisii | LMG 26277[T] | MLFS00000000.1 | Palmer et al. (2017) |
| | Pantoea sp. | At-9b | CP002433.1, CP002434.1, CP002435.1, CP002436.1, CP002437.1, CP002438.1 | Suen et al. (2010) |
| | Pantoea sp. | A4 | ALXE00000000.1 | Hong et al. (2012) |
| | Pantoea sp. | GM01 | AKUI00000000.1 | Brown et al. (2012) |
| Tatumella | Tatumella morbirosei | LMG 23360[T] | CM003276.1 | – |
| | Tatumella ptyseos | ATCC 33301[T] | ATMJ00000000.1 | – |
| | Tatumella saanichensis | NML 06-3099[T] | ATMI00000000.1 | Tracz et al. (2015) |
| Erwinia | Erwinia billingiae | NCPPB 661[T] | FP236843.1, FP236826.1, FP236830.1 | Kube et al. (2010) |
| | Erwinia pyrifoliae | DSM 12163[T] | FN392235.1, FN392236.1, FN392237.1 | Kube et al. (2010) |
| | Erwinia tasmaniensis | Et 1-99[T] | CU468135.1, CU468128.1, CU468130.1, CU468131.1, CU468132.1, CU468133.1 | Kube et al. (2008) |

**Notes.**
[a] Superscript[T] indicates type strains for the species.
[b] All numbers refer to GenBank assembly accession numbers (http://www.ncbi.nlm.nih.gov/; accessed 28/2/2017).

subjecting the unrooted AML phylogenies to an MSC analysis in ASTRAL v. 5.6.3 (*Mirarab et al., 2014*). Outputs were indicated with branch lengths in coalescent units and support values for the four clusters around a specific branch (quartet score; *Sayyari & Mirarab, 2016*). Additionally, three other approaches were used to infer the *Pantoea* species tree. The first involved inference of a Neighbour-Joining (NJ) tree using distances based on Average Nucleotide Identity (ANI; *Richter & Rosselló-Móra, 2009*) values. These were available from a previous study (*Palmer et al., 2017*) and used to generate a pairwise distance matrix in Microsoft Excel™ from which a NJ tree was inferred using MEGA v. 6.06 (*Tamura et al., 2013*). Note that this precluded resampling of the data for evaluating branch support. Secondly, a Neighbor-Net network was inferred from the concatenated nucleotide data using default settings in SplitsTree v. 4 (*Huson & Bryant, 2005*). Thirdly, this software was also used to construct a consensus network from the single gene AML phylogenies with a zero threshold (exclude no trees) and edge weights set to count.

To determine the degree of congruence and distribution of signal across individual gene genealogies relative to the *Pantoea* species phylogenies, individual phylograms were manually inspected. During this process, gene genealogies supporting specific backbone nodes (that were consistently recovered using multiple inference approaches) within the *Pantoea* species phylogenies were identified. This was done by evaluating a set of twelve query hypotheses (representing all of the internal backbone nodes in the *Pantoea* species trees) against each of the individual gene genealogies to determine whether they contained and/or supported the expected nodes. Each genealogy was then marked as (1) fully supporting, (2) supporting, but with other taxa nested, (3) not supporting or (4) lacking signal for the specific node depicted in the query hypothesis. The signal obtained from each of the different gene trees were then related back to the physical order of the shared genes as they appear in the genome of *P. agglomerans* (*Lim et al., 2014*), to determine whether specific signal patterns could be associated to areas of the genome.

As an indication of how phylogenetic conflicts were distributed across the concatenated nucleotide alignment, incongruent signals for the *P. dispersa* and *P. ananatis* lineages were investigated. For these purposes, the nucleotide sites causing incongruence in the Neighbor-Net network was noted and related back to the gene identifier. These data were visualised across the concatenated alignment using Circos v. 0.69 (*Krzywinski et al., 2009*).

## Recombination detection

To determine whether recombination, as an organism-level evolutionary process, could have contributed to phylogenetic conflict within the dataset, genes with possible signals for recombination were identified. This was done by subjecting the concatenated data matrix to the Recombination Detection Program (RDP) v. 4.84 (*Martin et al., 2015*), to test for recombination breakpoints using five genetic distance-based methods (RDP, GENECONV, MaxChi, Chimaera and 3Seq). RDP employs a sliding window to calculate pairwise distances between all unique taxon triplets for parsimony informative sites. Regions in contradiction to a UPGMA dendrogram, constructed from all sites, are identified as potentially recombinant (*Martin & Rybicki, 2000*). The GENECONV method entails pairwise comparisons of all polymorphic sites within the alignment to identify

higher than expected similarity over unusually long regions compared to the rest of the alignment (*Padidam, Sawyer & Fauquet, 1999*). MaxChi identifies potential recombination breakpoints by examining differences in the proportions of variable polymorphic sites using a sliding window to calculate pairwise $\chi^2$ values (*Smith, 1992*). The Chimaera approach is in essence a modification of the MaxChi method, where triplets are screened using a sliding window for only polymorphic sites where recombinants match one of the parental sequences (*Posada & Crandall, 2001*). Lastly, the 3Seq method uses the same character set as Chimaera to query each sequence within each triplet combination to determine if it could be a possible recombinant of the other two sequences (*Boni, Posada & Feldman, 2007*). These data were also plotted on the concatenated alignment using Circos v. 0.69 (*Krzywinski et al., 2009*).

### Randomised subset phylogenetic analyses

To determine whether limited sets of genes contained sufficient phylogenetic signal to overcome phylogenetic conflict within the dataset, randomised subsets of 20, 50, 60, 70, 80, 90, 100, 110 and 120 genes were constructed. For this purpose, genes were randomly identified in Microsoft Excel™ [=RANDBETWEEN(1,1357)] without resampling. The concatenation and phylogenetic analyses were conducted in the same manner as described above. In all cases, ten individual data subsets were constructed, followed by obtaining a strict and majority rule consensus tree of the ten phylograms of each gene set (i.e., 20, 50, 60, 70, 80, 90, 100, 110 or 120 genes).

## RESULTS

### Detecting phylogenetic conflict

Using the AML approach, a robust and well-supported evolutionary hypothesis regarding the species relationships in *Pantoea* was obtained. The AML phylogeny was based on 337,780 amino acid columns corresponding to the protein sequences of 1,357 genes (Fig. 1). This phylogeny was also congruent with the phylogeny obtained with a larger taxon set for *Pantoea*, *Erwinia*, *Tatumella* and outgroup taxa by *Palmer et al. (2017)*, where ML inferences were performed with the appropriate evolutionary models for each gene partition. The only exceptions were the sister-grouping between *P. agglomerans* and *P. vagans* in the current tree, however due to their close relatedness this is not an uncommon problem, and the grouping of *P. deleyi* and *P. anthophila*. Similarly, a robust and equally well-supported phylogeny was obtained using the MSC approach where the species tree was inferred from the set of 1,357 individual gene trees (Fig. 1 and Fig. S1). Overall, the MSC topology was also congruent with the phylogeny obtained by *Palmer et al. (2017)*. Exceptions were only observed at nodes at tips or leave nodes (i.e., the groupings observed in the *P. agglomerans* and *P. dispersa* lineages).

   The AML and MSC topologies were highly congruent (Fig. 1). The only differences were those regarding relationships within the *P. dispersa* lineage and the *P. agglomerans* lineage (both topologies also lacked support for the relationships within this lineage). In terms of the MSC topology, comparison of the quartet scores for the main, the first alternative and second alternative topologies possible at each node (*Sayyari & Mirarab, 2016*), showed that
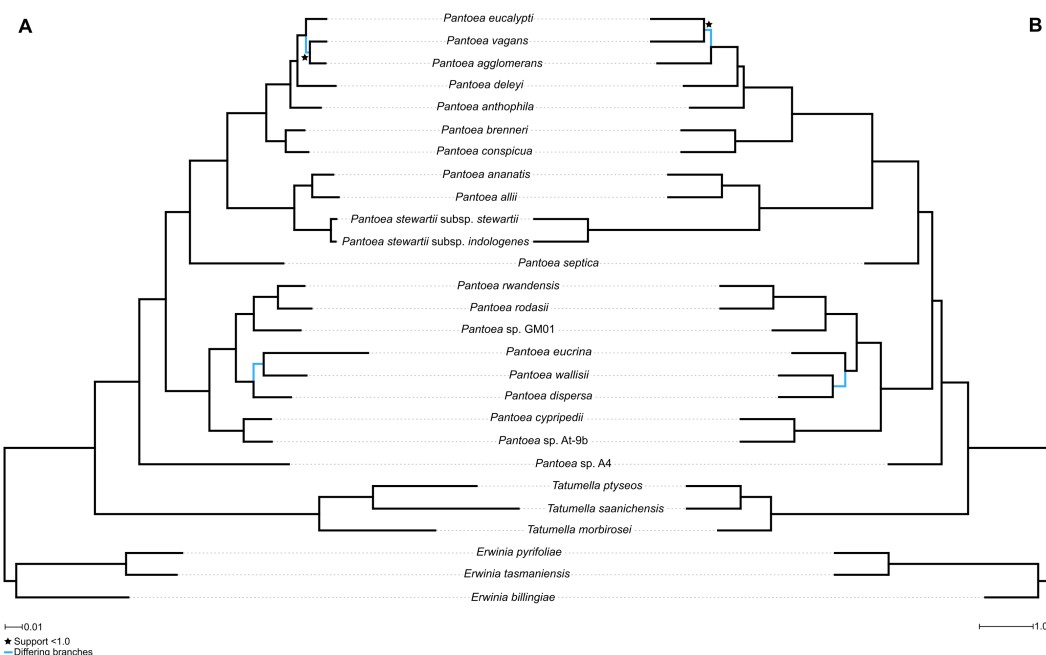

**Figure 1** **Comparison between the AML and MSCM phylogenies.** Blue branches indicate differences in topology and branches with support of lower than 1.0 are indicated with a star. (A) The approximate maximum likelihood (AML) phylogeny constructed from the concatenated data matrix of the protein sequences of 1,357 genes, consisting of 337,780 amino acid columns. The phylogeny was constructed with FastTree v. 2.1 (*Price, Dehal & Arkin, 2010*) with the JTT (*Jones, Taylor & Thornton, 1992*) evolutionary model with CAT approximation. Simodaira-Hasegawa branch support values from 1,000 replicates were used. (B) A species phylogeny using the multispecies coalescent model as implemented in ASTRAL v.5.6.3 (*Mirarab et al., 2014*) based on the individual phylogenies constructed from the protein sequences of 1,357 genes. The scale bar indicates one coalescent unit (*Mirarab et al., 2014*). Terminal branches are indicated as one coalescent unit, as branch lengths for taxa corresponding to species can only be calculated where multiple individuals per species are analysed. Shorter branches correspond to higher levels of incongruence and are generally associated with high levels of incomplete lineage sorting (ILS). Support values are determined based on Bayesian posterior probability values computed from the single gene tree quartet frequencies (*Sayyari & Mirarab, 2016*).

the nodes where quartet scores between the topologies differed very little (where quartet scores for the three alternatives were almost equal) were generally those responsible for incongruence between the topologies inferred using different approaches (Fig. 1 and Fig. S1; particularly within the *P. agglomerans* and *P. dispersa* lineages). This suggests that none of these approaches are particularly robust when resolving closely related or undersampled lineages close to the leaves of the phylogenies.

The network approaches indicated a large amount of conflicting signal within the data. This was evident in the Neighbor-Net network (Fig. 2 and File S2) based on the concatenated nucleotide data matrix (1,010,946 bases), as well as the Consensus Network (Fig. S2 and File S3) of the individual gene trees (1,357 protein sequences). These conflicting signals were particularly prevalent at the deeper edges of the evolutionary hypotheses, e.g., the *P. dispersa* lineage compared to the *P. ananatis* lineage (denoted A and B in Fig. 1). However, despite

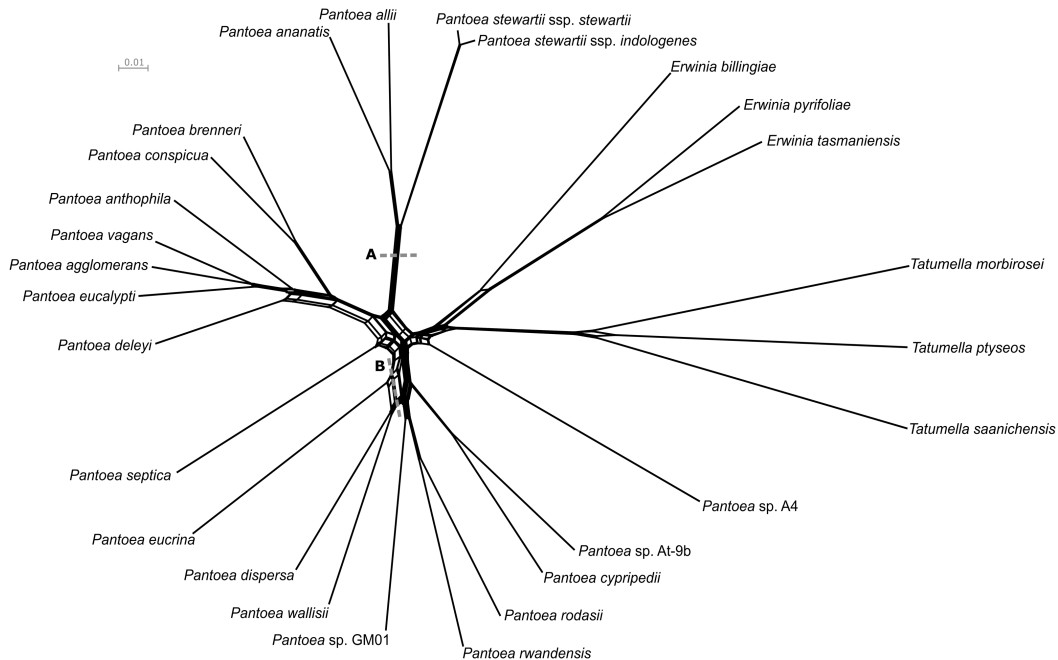

**Figure 2  Neighbor-Net network from the concatenated nucleotide data.** The Neighbor-Net network was constructed from *p*-distances with equal angles for the concatenated nucleotide dataset. Overall, the configuration of the network is congruent with the existing species phylogeny for the genus *Pantoea*. Clear separation between the *P. agglomerans* and *P. ananatis* lineages were obtained and were also clearly distinct from the *P. rodasii* and *P. dispersa* lineages. Point A denotes where signal in conflict to the grouping of the *P. ananatis* lineage was determined, while point B denotes where signal in conflict to the grouping of the *P. dispersa* lineage was determined (see text for details).

the presence of this conflict, the evolutionary hypotheses obtained with the networks, overall, reflected the relationships obtained for the AML and MSC phylogenies (Figs. 1 and 2 and Fig. S2). Furthermore, the topology obtained for the ANI-based distances was mostly congruent to the lineages recovered from the various species tree inference approaches (Fig. S3). All backbone nodes that were consistently recovered in the other approaches, were also recovered with the ANI-based distances with the exception of *P. eucrina* grouping as sister to the singleton, *P. septica*.

To determine the degree of incongruence caused by phylogenetic conflict, comparisons of all individual gene trees were evaluated against a set of twelve query phylogenetic hypotheses (Fig. 3A). These query hypotheses were constructed to evaluate monophyly of lineages or groups across the backbone of the *Pantoea* species phylogenies, thus shallower nodes near the tips of the trees (leaves) were not considered. None of the 1,357 gene trees were fully congruent with the respective phylogenetic hypotheses of *Pantoea*. Of the individual gene trees, only six genes supported all the nodes in the backbone for the groupings observed previously (File S4). Additionally, seven gene genealogies produced polytomies of taxa and thus were marked as containing no signal for any of the nodes observed in the phylogenetic hypotheses of *Pantoea* (Fig. 4 and File S4). The remaining gene trees supported at least one of the nodes in the backbone observed in the *Pantoea* species trees. Exclusion from

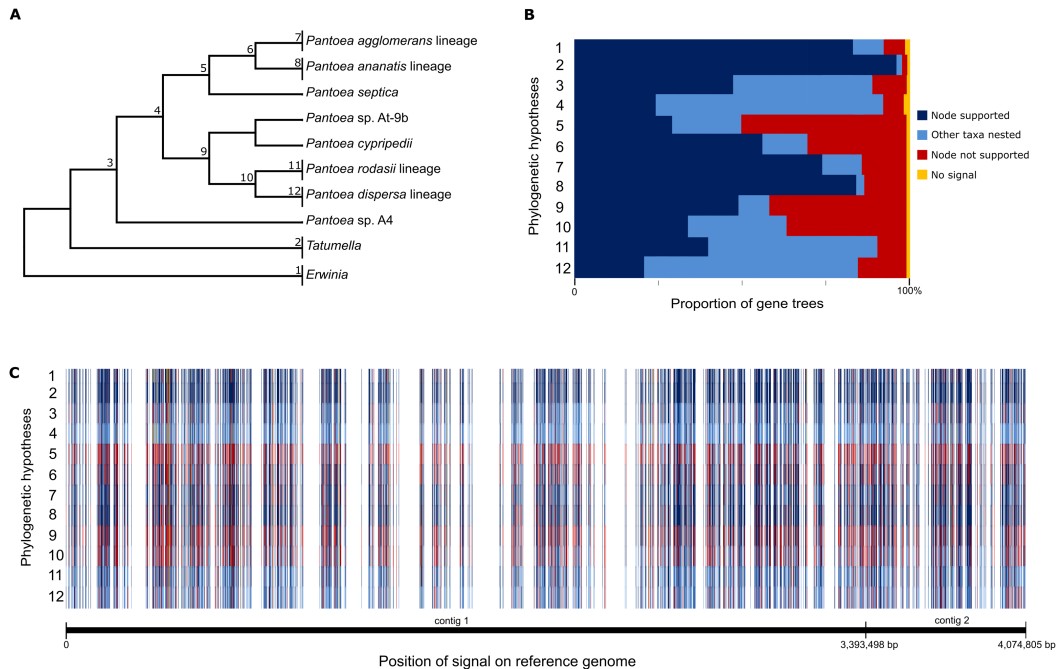

**Figure 3** **The summary of individual gene tree comparisons.** (A) The phylogenetic hypotheses evaluated during topology comparisons. Each number represents a specific hypothesis, where the monophyly of the group at each node was evaluated. An example is hypothesis 3, where the overall monophyly of *Pantoea* was evaluated. (B) A relative frequency histogram depicting the proportion of individual gene trees that support the phylogenetic hypotheses evaluated. Dark blue indicates genes that fully supported the monophyly of the corresponding hypothesis, while light blue indicates support for the monophyly of the hypothesis with additional taxa nested within the group. Red indicates gene trees that were incongruent with the corresponding hypotheses and yellow denotes gene trees with no signal (polytomies). (C) The signal obtained for each gene genealogy compared to the phylogenetic hypotheses were plotted against the position of the genes on the chromosome of *Pantoea agglomerans*. The same colour scheme is applied as in the frequency histogram. All genes were located on the chromosome of *P. agglomerans* R190 and was distributed across the chromosome consisting of two contigs. Signal for the respective nodes within the species phylogeny were distributed across the chromosome and no patterns of shared signal were detected for groups of adjacent genes.

the concatenated analyses of either the six backbone-supporting genes or the seven genes providing no resolution among taxa, still provided the same overall topology, with the exception of the grouping of *P. agglomerans*, *P. vagans* and *P. eucalypti* (Figs. S4A and S4B) that lacked statistical support. The phylogenies constructed from the concatenated datasets with only the six backbone-supporting genes and only the seven genes showing no signal (confirmed with RAxML v. 8.0.20; Fig. 3H), also allowed the recovery of a mostly congruent phylogeny to that of the expected topology, but with very low or no support at a number of nodes and slight interspecies differences in the *P. agglomerans* lineage and the position of singleton taxa (Fig. 3H, Figs. S5A and S5B).

Based on these topology comparisons, it appeared that the signal supporting different nodes across the *Pantoea* species phylogenies were supported by different genes. As a means to investigate the distribution of phylogenetic conflict at the gene partition-level across the

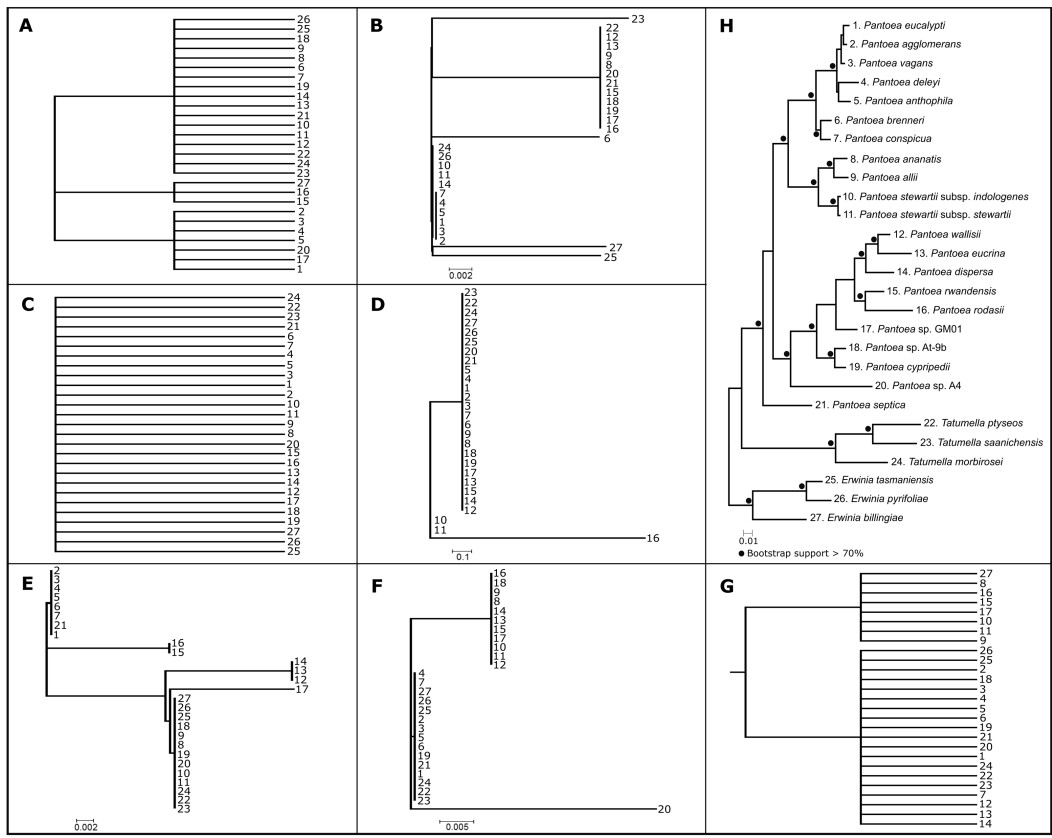

**Figure 4** **Summary of genes with limited to no signal.** Seven single gene phylogenies determined with approximate maximum likelihood (AML) analyses for genes identified as containing no signal (see File S4) and the maximum likelihood (ML) phylogeny inferred from the combined sequence of these seven genes. (A–G) The AML phylogeny constructed from the protein sequences for the 30S ribosomal protein S18, UDP-diphospho-muramoylpentapeptide beta-N-acetylglucosaminyltransferase, Prolyl-tRNA synthetase, Glutamate 5-kinase, Cold shock-like protein *cspC*, 30S ribosomal protein S10 and 30S ribosomal protein S12, respectively. Taxa are numbered according to taxon descriptors in H. (H) The concatenated ML phylogeny constructed using RAxML v. 8.0.20 with the appropriate amino acid model inferred using ProtTest v. 3.4 for each partition. All bootstrap support values above 70% are indicated at nodes with dots. The phylogeny resembles the known species phylogeny for *Pantoea* with the exception of some species relationships within the *P. agglomerans* and *P. rodasii* lineages and the grouping of singleton taxa.

genome, the signal for each gene was plotted against the genome of *P. agglomerans* (Fig. 3C). All shared genes were localized to the chromosome of *P. agglomerans*. This analysis also revealed that signal for all nodes were randomly distributed across the chromosome of *P. agglomerans* and no apparent patterns of shared signal were detected for adjacent genes (Fig. 3C).

To interrogate the distribution of conflict across the dataset at the nucleotide-level, nucleotide positions in phylogenetic conflict with relationships observed within the *Pantoea* species phylogeny were identified. Of the 1,010,946 bases within the nucleotide alignment, 493,834 bases (48.7%) were identical across all taxa, with 517,112 nucleotide positions being variable between taxa. For these analyses the *P. ananatis* lineage, with the

least conflicting signal within the dataset (Fig. 2A), and the *P. dispersa* lineage, with the most conflicting signal within the dataset (Fig. 2B), were investigated as a best and worst case scenario, respectively. For the *P. ananatis* and *P. dispersa* lineages, a total of 1,764 and 3,856 nucleotide sites, respectively, supported relationships differing from the *Pantoea* species phylogenies (Fig. 2 points A and B; File S5). However, these sites were distributed across the concatenated alignment and were not localized to specific genomic regions (Fig. 5).

## Recombination as a source of phylogenetic conflict

Using RDP, a total of 276 potential recombination events were detected (File S6), with 166 of these indicated as potentially caused by evolutionary processes other than recombination (*Martin et al., 2015*). This yielded 110 likely recombination events, occurring across 54 regions of the concatenated sequence, supported by at least three different analytical methods, of which 57 events were supported by all five methods (File  S6). However, to avoid the inclusion of potentially artefactual recombination breakpoints associated with the concatenation process, we only considered recombination breakpoints occurring within the boundaries of single genes. This yielded a total of 15 recombination events, identified across 11 genes within the concatenated alignment (Fig. 5 and File S6). From these results, recombination break-points were detected in members of all lineages within *Pantoea*. None of these recombination breakpoints could, however, be linked to the nucleotide-level phylogenetic conflict identified (File S5).

## Phylograms from limited sets of genes

The topology of the *Pantoea* species phylogenies could be recovered by some of the randomised subsets of 20, 50 and 60 genes, whereas all subsets containing the information for 70 or more genes recovered these nodes. Within each set of ten replicate data subsets, the length of individual alignments differed depending on the length of the specific genes used to construct them (Table 2 and File S7). For example, for the 20 gene subsets, the lengths of the alignments ranged from 5,560 to 7,152 amino acid columns, while the length of the alignments for the 120 gene subsets ranged from 36,614 to 42,793 amino acid columns (Table 2).

Overall, support for the backbone of the *Pantoea* species tree (Fig. 3A) deteriorated with a decrease in the number of genes concatenated and analysed (Table 2). When fewer genes were analysed in the replicates, various branches collapsed and branch support decreased in the strict consensus trees (Table 2, Fig. 6 and Fig. S6). Overall, strict consensus trees from the various replicates of 70 and more genes resulted in the recovery of a phylogeny congruent with the species phylogenies of *Pantoea*, however multiple individual replicates of the smaller datasets produced trees that were largely incongruent with these hypotheses. Only the trees from multi-gene subsets of 70 or more genes, consistently allowed robust and well-supported reconstruction of the expected *Pantoea* species trees, specifically with regards to branches in the backbone of the phylogeny (i.e., query hypotheses; Fig. 6).
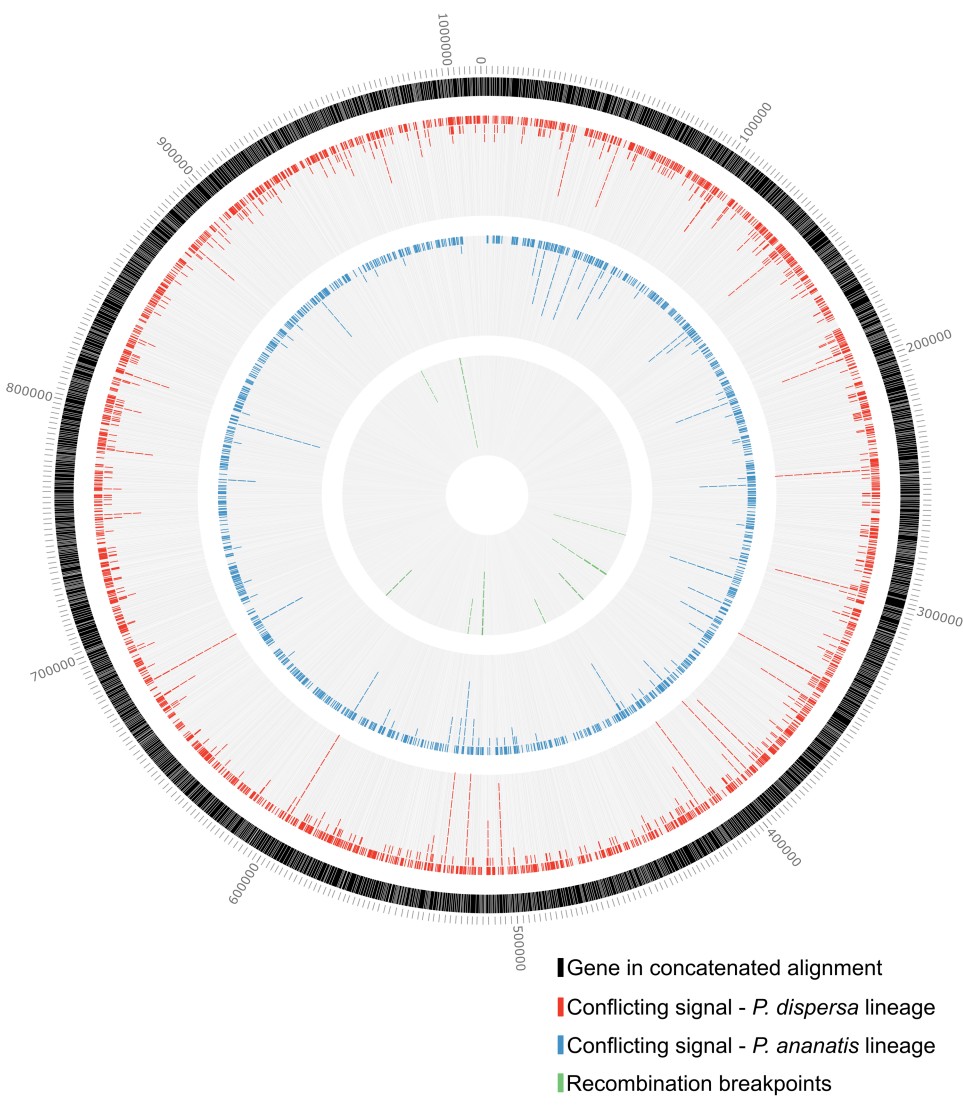

■ Gene in concatenated alignment

■ Conflicting signal - *P. dispersa* lineage

■ Conflicting signal - *P. ananatis* lineage

■ Recombination breakpoints

**Figure 5  Conflicting signal and possible recombination breakpoints.** A circular diagram depicting the nucleotide concatenated alignment of all shared genes. The outer track indicates the gene boundaries within the alignment, with tick marks representing the length in nucleotides at 2,000 bp intervals. The second track indicates the nucleotide positions within genes supporting conflicting topologies for the *P. dispersa* lineage to species groupings observed in the concatenated species phylogeny. A total of 3,856 nucleotide positions supported conflicting topologies for the *P. dispersa* lineage. The third track indicates nucleotide positions supporting conflicting topologies for the *P. ananatis* lineage compared to species groupings observed in the concatenated species phylogeny. For this lineage 1,764 nucleotide positions supported conflicting topologies for the *P. ananatis* lineage. The inner track represents recombination breakpoints detected within gene boundaries for the concatenated alignment. These breakpoints were supported by at least three of the five methods employed (RDP, GENECONV, Chimaera, MaxChi and 3Seq) for detecting recombination. Stacked tiles reflect the number of methods that were successful in detecting recombination events at those regions, as well as multiple recombination events within the same region in various species (See File  S6).

**Table 2   Summary of gene subset tests[a].**

| Number of genes in subset | Replicate | Length (bp) | Backbone nodes support range[b] | Leaf nodes support range |
|---|---|---|---|---|
| 120 genes | 1 | 40,069 | 1.00 | 0.50–1.00 |
| | 2 | 36,614 | 1.00 | 0.37–1.00 |
| | 3 | 39,663 | 1.00 | 0.93–1.00 |
| | 4 | 38,050 | 1.00 | 0.43–1.00 |
| | 5 | 37,931 | 1.00 | 0.46–1.00 |
| | 6 | 40,260 | 1.00 | 0.57–1.00 |
| | 7 | 39,776 | 1.00 | 0.82–1.00 |
| | 8 | 40,385 | 1.00 | 0.86–1.00 |
| | 9 | 42,793 | 1.00 | 0.67–1.00 |
| | 10 | 39,328 | 1.00 | 0.59–1.00 |
| 110 genes | 1 | 35,298 | 1.00 | 0.95–1.00 |
| | 2 | 35,349 | 1.00 | 0.69–1.00 |
| | 3 | 36,798 | 1.00 | 0.86–1.00 |
| | 4 | 38,800 | 1.00 | 0.40–1.00 |
| | 5 | 35,445 | 1.00 | 0.73–1.00 |
| | 6 | 38,042 | 1.00 | 0.71–1.00 |
| | 7 | 40,172 | 0.99–1.00 | 0.38–1.00 |
| | 8 | 39,865 | 1.00 | 0.18–1.00 |
| | 9 | 40,737 | 1.00 | 0.78–1.00 |
| | 10 | 40,745 | 1.00 | 0.54–1.00 |
| 100 genes | 1 | 33,340 | 1.00 | 0.78–1.00 |
| | 2 | 30,822 | 0.99–1.00 | 0.47–1.00 |
| | 3 | 33,433 | 1.00 | 0.65–1.00 |
| | 4 | 30,707 | 1.00 | 0.90–1.00 |
| | 5 | 31,340 | 1.00 | 0.58–1.00 |
| | 6 | 31,798 | 1.00 | 0.87–1.00 |
| | 7 | 29.562 | 1.00 | 0.64–1.00 |
| | 8 | 30,773 | 1.00 | 0.06–1.00 |
| | 9 | 34,064 | 1.00 | 0.88–1.00 |
| | 10 | 35,550 | 1.00 | 0.68–1.00 |
| 90 genes | 1 | 31,353 | 1.00 | 0.68–1.00 |
| | 2 | 29,307 | 0.99–1.00 | 0.91–1.00 |
| | 3 | 31,941 | 1.00 | 0.94–1.00 |
| | 4 | 31,890 | 1.00 | 0.77–1.00 |
| | 5 | 29,695 | 1.00 | 0.84–1.00 |
| | 6 | 30,564 | 1.00 | 0.37–1.00 |
| | 7 | 25,162 | 1.00 | 0.78–1.00 |
| | 8 | 30,745 | 1.00 | 0.48–1.00 |
| | 9 | 28,146 | 1.00 | 0.55–1.00 |
| | 10 | 28,883 | 1.00 | 0.81–1.00 |

**Table 2** (*continued*)

| Number of genes in subset | Replicate | Length (bp) | Backbone nodes support range[b] | Leaf nodes support range |
|---|---|---|---|---|
| 80 genes | 1 | 24,020 | 1.00 | 0.35–1.00 |
| | 2 | 23,065 | 1.00 | 0.73–1.00 |
| | 3 | 25,922 | 0.99–1.00 | 0.86–1.00 |
| | 4 | 27,877 | 1.00 | 0.53–1.00 |
| | 5 | 25,288 | 0.99–1.00 | 0.70–1.00 |
| | 6 | 22,551 | 0.98–1.00 | 0.69–1.00 |
| | 7 | 26,417 | 1.00 | 0.59–1.00 |
| | 8 | 27,008 | 1.00 | 0.72–1.00 |
| | 9 | 25,156 | 1.00 | 0.30–1.00 |
| | 10 | 25,498 | 0.98–1.00 | 0.77–1.00 |
| 70 genes | 1 | 22,011 | 0.99–1.00 | 0.16–1.00 |
| | 2 | 24,373 | 1.00 | 0.54–1.00 |
| | 3 | 24,420 | 1.00 | 0.45–1.00 |
| | 4 | 20,887 | 1.00 | 0.82–1.00 |
| | 5 | 22,286 | 0.99–1.00 | 0.11–1.00 |
| | 6 | 22,702 | 1.00 | 0.27–1.00 |
| | 7 | 23,787 | 0.99–1.00 | 0.83–1.00 |
| | 8 | 19,750 | 1.00 | 0.21–1.00 |
| | 9 | 23,770 | 0.99–1.00 | 0.68–1.00 |
| | 10 | 21,613 | 1.00 | 0.31–1.00 |
| 60 genes | 1 | 18,755 | 0.99 - 1.00 | 0.92–1.00 |
| | 2 | 21,310 | 0.99–1.00 | 0.89–1.00 |
| | 3 | 21,745 | 0.99–1.00 | 0.77–1.00 |
| | 4 | 19,210 | 1.00 | 0.83–1.00 |
| | 5 | 19,495 | 1.00 | 0.83–1.00 |
| | 6 | 18,550 | 0.83 - 1.00 | 0.69–1.00 |
| | 7 | 20,389 | 1.00 | 0.58–1.00 |
| | 8 | 20,475 | 0.77–1.00 | 0.47–1.00 |
| | 9 | 17,331 | 1.00 | 0.01–1.00 |
| | 10 | 23,324 | 0.97–1.00 | 0.40–1.00 |
| 50 genes | 1 | 14,890 | 1.00[*] | 0.31–1.00 |
| | 2 | 18,079 | 1.00 | 0.27–1.00 |
| | 3 | 14,701 | 1.00 | 0.81–1.00 |
| | 4 | 13,983 | 0.71–1.00 | 0.00–1.00 |
| | 5 | 19,059 | 1.00 | 0.40–1.00 |
| | 6 | 18,412 | 0.99–1.00 | 0.86–1.00 |
| | 7 | 18,880 | 1.00 | 0.59–1.00 |
| | 8 | 14,411 | 0.85–1.00 | 0.33–1.00 |
| | 9 | 14,942 | 0.81–1.00 | 0.67–1.00 |
| | 10 | 14,531 | 1.00 | 0.28–1.00 |

**Table 2** (*continued*)

| Number of genes in subset | Replicate | Length (bp) | Backbone nodes support range[b] | Leaf nodes support range |
|---|---|---|---|---|
| 20 genes | 1 | 5,966 | 0.87–1.00 | 0.74–1.00 |
| | 2 | 5,834 | 0.99–1.00[*] | 0.64–1.00 |
| | 3 | 6,859 | 0.98–1.00 | 0.58–1.00 |
| | 4 | 7,152 | 0.97–1.00[*] | 0.00–1.00 |
| | 5 | 5,560 | 0.99–1.00 | 0.26–1.00 |
| | 6 | 6,316 | 0.48–1.00 | 0.71–1.00 |
| | 7 | 6,210 | 0.93–1.00 | 0.26–1.00 |
| | 8 | 6,517 | 0.99–1.00 | 0.22–1.00 |
| | 9 | 6,649 | 0.95–1.00 | 0.06–1.00 |
| | 10 | 6,436 | 0.90–1.00 | 0.53–1.00 |

**Notes.**
[a] See File S3.
[b] One backbone node is not recovered in the phylogenies marked with an asterisk.

# DISCUSSION

This study employed a novel approach to investigate phylogenetic conflict within concatenated datasets. We interrogated the distribution and effect of both phylogenetic signal and conflict at the gene partition and nucleotide levels. This entailed the use of various phylogenetic analyses coupled with manual inspection and evaluation of individual gene trees. These data revealed the effects of phylogenetic conflict and signal in concatenated datasets, which are the input typically used for phylogenomic reconstruction. Our findings support the idea that all genes, even if they appear to be phylogenetically uninformative when analysed alone, contribute signal toward a phylogenomic evolutionary hypothesis and that the obtained topology is not driven by single genes. This is reminiscent of Aristotle's idea of synergism that "the whole is greater than the sum of its parts". In other words, by concatenating single genes a synergistic effect is achieved, where the combined data seems to be superior to that of the proverbial sum of the signal.

As demonstrated previously (*Palmer et al., 2017*), the full set of shared genes allowed reconstruction of a robustly supported phylogenetic hypothesis for *Pantoea* using AML. In fact, it is quite common to obtain a robust, highly supported phylogeny through concatenation of all shared genes, despite the incongruent nature of individual gene trees (*Hedtke, Townsend & Hillis, 2006*; *Jeffroy et al., 2006*; *Rokas et al., 2003*; *Salichos & Rokas, 2013*; *Thiergart, Landan & Martin, 2014*). However, none of our single gene genealogies were fully congruent with the phylogenomic species tree of *Pantoea*, while only six genes allowed recovery of the backbone of the species phylogeny. This was not surprising as various previous studies showed that very few or no genes typically support a particular species phylogeny fully (*Dagan & Martin, 2006*; *Hedtke, Townsend & Hillis, 2006*; *Jeffroy et al., 2006*; *Rokas et al., 2003*; *Salichos & Rokas, 2013*; *Thiergart, Landan & Martin, 2014*). In contrast to conclusions drawn previously (*Thiergart, Landan & Martin, 2014*), most of the *Pantoea* gene trees supported at least some of the nodes within the species phylogeny. In other words, support for the respective nodes was not necessarily obtained from the same genes but rather scattered across different genes.

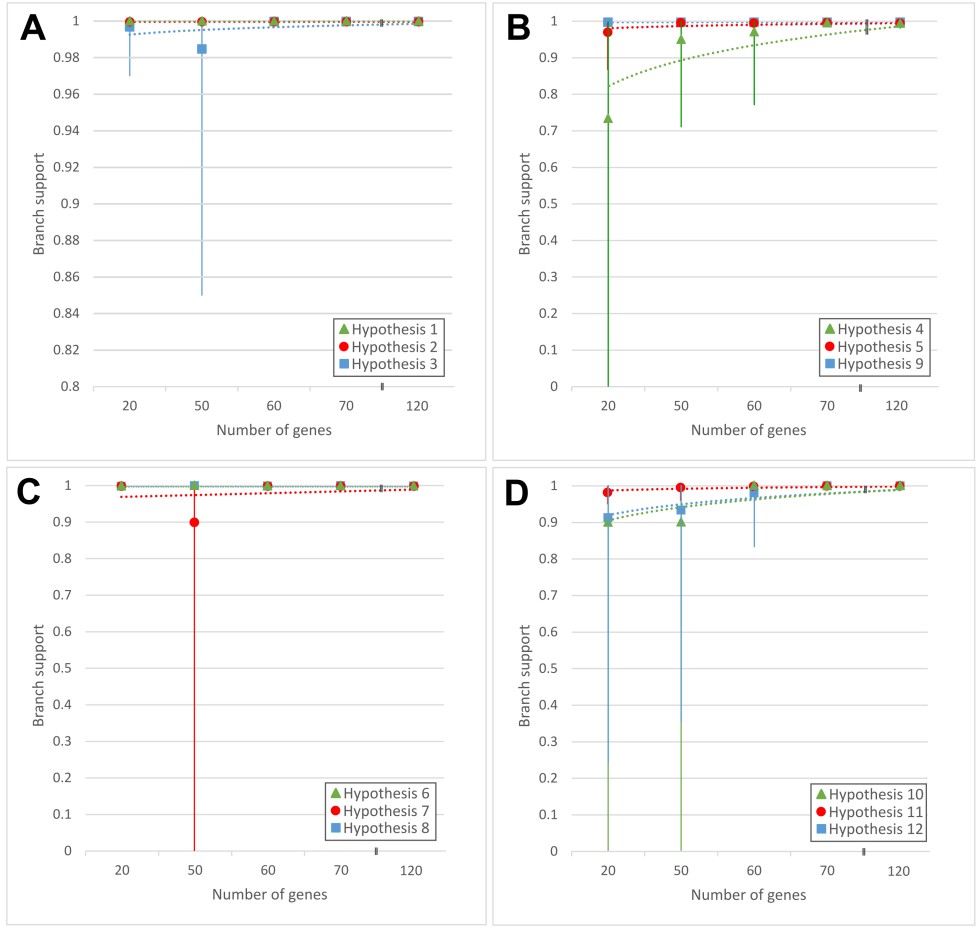

**Figure 6** **The SH branch support for specific hypotheses in the trees constructed from the subset datasets.** Each hypothesis (Fig. 3A) was interrogated in each of the subset tree datasets, where 20, 50, 60, 70, 80, 90, 100, 110 and 120 genes were used to construct ten randomised datasets for each number of genes (File S7). The range indicated for each data point stretches from the lowest branch support (0 in the case where the nodes were not recovered) to the highest branch support (1 where the branch was fully supported) with the mean indicated with the data point. Regression analyses were performed in Microsoft Excel™ to fit the best regression model to the data. (A) Support for the three hypotheses depicted represent the monophyly of the three genera *Erwinia* (hypothesis 1; green), *Tatumella* (hypothesis 2; red) and *Pantoea* (hypothesis 3; blue). All subsets datasets recovered the nodes representing the monophyly of the genera, but in the case of *Pantoea*, with less support in the replicates of the lower number of genes. (B) The support for the three test hypotheses (i) separating the remainder of *Pantoea* from *Pantoea* sp. A4 (hypothesis 4; green), (ii) grouping *P. septica*, and the *P. agglomerans* and *P. ananatis* lineages together (hypothesis 5; red) and (iii) the grouping of *P. cypripedii* and *Pantoea* sp. At-9b with the *P. rodasii* and *P. dispersa* lineages (hypothesis 9; blue). The node representing hypothesis 4 were not recovered in two repeats of the 20 gene subsets. (C) Hypothesis 6 depicts the sister grouping of the *P. agglomerans* and *P. ananatis* lineages with the support associated with the node depicted in green. The monophyletic grouping of the *P. agglomerans* lineage (hypothesis 7; red) were not recovered in one 50 gene repeat, but were further fully supported in all repeats. The grouping of the *P. ananatis* lineage was consistently recovered with full support (hypothesis 8; blue). (D) The support associated with the nodes depicting the sister relationship between the *P. rodasii* and *P. dispersa* lineages (hypothesis 10; green). This node was not recovered in one of the 20 gene repeats and one of the 50 gene repeats. hypothesis 11 represents the monophyletic grouping of the *P. rodasii* lineage, which was consistently recovered and well-supported with branch support >0.95 (red). Branch support associated with the monophyletic grouping of the *P. dispersa* lineage often ranged from very low (0.24) to fully supported (1) in the 20, 50 and 60 gene repeats (hypothesis 12; blue).

A fully resolved, well-supported phylogeny was obtained using the MSC approach, although it was not congruent with the AML tree regarding relationships at the tips of the trees. Concatenation is thought to be superior to MSC-based species trees if ILS is low, while MSC models perform better in the presence of moderate ILS (*Mirarab et al., 2014*). As our MSC and AML trees correspond perfectly regarding backbone nodes, these are strong hypotheses that may approach the real relationships among these taxa. However, because organism-level evolutionary processes were not quantified in this study, we cannot exclude the possibility that ILS were responsible for the incongruences observed. Our study therefore highlights that alternative approaches that model genome evolution, quantify organism-level evolutionary processes (*Szöllősi et al., 2012*; *Williams et al., 2017*), and that focusses on leave taxa are needed to fully resolve the species tree of a diverse assemblage such as *Pantoea*. This may be particularly true when taxa are very closely related or when the lineages in question are undersampled.

The random distribution of signal across the *Pantoea* genome is supported by the recovery of overall congruent subset phylogenies from random sub-samplings of gene sequences. Due to this random distribution, one should be able to obtain sufficient signal to reconstruct the species phylogeny by randomly sampling enough genes from the genome (*Dutilh et al., 2004*; *Gadagkar, Rosenberg & Kumar, 2005*). From our data, this idea was tested with consensus trees of 10 replicates with 20, 50, 60, 70, 80, 90, 100, 110 and 120 genes. We found that, with a decrease in the number of genes analysed, support for the backbone and the deeper branches decreased incrementally, as has also been observed previously (*Rokas et al., 2003*). Therefore, for these data and taxon set, it appear that at least 70 randomly selected genes from the genome is required to obtain a relatively robust, well-supported phylogeny, particularly to reconstruct the deeper relationships within and among the genera. Multi-gene phylogenies based on 70 genes may thus provide sufficiently robust hypotheses so that complete genome sequence data may not be required, our work suggests that sufficient data may be obtained from low level sequencing, although verification of this notion in other taxon sets is required.

The species tree hypotheses for *Pantoea* was generally also supported by the two network approaches employed here. Both accommodated non-vertical and non-phylogenetic signal (introduced through systematic error) as inferred from nucleotide data, as well as the individual gene trees (*Bryant & Moulton, 2002*; *Holland & Moulton, 2003*; *Holland, Jermiin & Moulton, 2005*). These methods produced networks in which the overall clustering patterns were generally congruent with that obtained through gene concatenation-based phylogenomic inferences. This would not have happened if insufficient signal (i.e., stochastic error *Jeffroy et al., 2006*; *Philippe et al., 2011*; *Rosenberg & Kumar, 2003*) or reconstruction artefacts (i.e., systematic error *Hedtke, Townsend & Hillis, 2006*; *Hillis, 1998*; *Pollock et al., 2002*; *Zwickl & Hillis, 2002*) were responsible for the observed relationships in the species trees. If conflict, particularly in the form of HGT and ILS, dominated the dataset, the splits graphs would not have such high overall congruence to the species trees (*Bryant & Moulton, 2002*), however, more in-depth future studies are required to fully elucidate the role and amount of organism-level evolutionary processes in the evolution of these taxa. Compared to previous analyses, often employing a limited gene set (*Chen et al.,*

*2013*; *Kennedy et al., 2005*), more box-like structures were observed in *Pantoea* networks, particularly in deeper edges. However, these boxes were generally smaller, where the increased number of boxes indicate more alternate or conflicting relationships, while the shorter edges correlate to the particular relationships being observed less frequently. If one considers that these conflicts are visualized for the full shared gene set, the level of conflict appears to be relatively low and comparable to that seen in other bacteria (*Retchless & Lawrence, 2010*), but considerably lower than taxa undergoing extensive HGT (*Doroghazi & Buckley, 2010*). The generally low level of conflict thus supports the idea that sufficient phylogenetic signal is present within the concatenated dataset to overshadow the limited conflict present.

Overall, the relationships obtained using the ANI-based distance approach were mostly congruent to the species trees. The incongruences that were present can be ascribed to the fact that ANI is notoriously unreliable as an indicator of relatedness, especially among more distantly related taxa (*Konstantinidis & Tiedje, 2007*; *Palmer et al., 2017*; *Qin et al., 2014*; *Rosselló-Mora, 2005*). This phenomenon is the reason why many prokaryotic taxonomist would rather purport the use of Average Amino Acid Identity (AAI) values at this level (*Konstantinidis & Tiedje, 2007*; *Qin et al., 2014*; *Rosselló-Mora, 2005*), as substitution saturation and other factors resulting from endogenous evolutionary processes may be responsible for the decline in informativeness of this metric, the more distantly related the taxa become (*Palmer et al., 2017*).

We investigated the phylogenetic conflict within the *Pantoea* dataset for the two lineages in which we observed the least and most phylogenetic conflict. Respectively, these were the *P. ananatis* lineage, which includes plant pathogenic species, and the *P. dispersa* lineage, which includes generalists (*Palmer et al., 2018*; *Palmer et al., 2017*; *Walterson & Stavrinides, 2015*). The number of nucleotide sites supporting alternate topologies to the species trees were limited, with only 0.75% of variable nucleotide sites (3,856 sites out of 517,112) supporting conflicting topologies in the *P. dispersa* lineage. Also, the remaining variable sites did not necessarily support the species relationships observed in the species trees, because different genes and nucleotide positions supported different nodes within the species phylogenies. Moreover, the conflicting signal within the dataset was not localised to specific genomic regions or genes, but rather, was randomly distributed. Taken together, these results thus suggest that (i) the use of network approaches for constructing phylogenies can be extremely valuable for identifying phylogenetic conflicts in datasets (*Bryant & Moulton, 2002*; *Holland & Moulton, 2003*; *Holland, Jermiin & Moulton, 2005*), and (ii) organism-level evolutionary processes like HGT and/or ILS impacts different lineages and taxa to varying degrees.

Conflicting phylogenetic signal in the *Pantoea* dataset could potentially result from recombination events that led to gene conversions between species (*Daubin, Moran & Ochman, 2003*; *Fraser, Hanage & Spratt, 2007*; *Holmes, Urwin & Maiden, 1999*; *Posada & Crandall, 2001*; *Posada & Crandall, 2002*). We found evidence for at least 15 recombination events in 11 shared genes in the dataset. We attributed these to recent instances of recombination, because older organism-level evolutionary events, particularly ancient HGT and ILS (*Knowles, 2009*; *Meng & Kubatko, 2009*; *Retchless & Lawrence, 2010*), become difficult to detect due to deterioration of signals by endogenous evolutionary

processes (*Daubin, Moran & Ochman, 2003*). It is also difficult to distinguish between these organism-level evolutionary processes as their signals may appear very similar (*Knowles, 2009*; *Nosil, 2008*; *Wendel & Doyle, 1998*) and future studies would be required to tease apart the roles of these processes in *Pantoea*. Nevertheless, identification of some of these organism-level evolutionary events in *Pantoea* provides possible mechanisms for how phylogenetic conflict could have been introduced into the data.

The use of all shared genomic information allowed for the recovery of robustly supported relationships, overcoming the weaknesses observed in individual gene datasets (*Andam & Gogarten, 2011*; *Daubin, Gouy & Perriere, 2002*; *Gadagkar, Rosenberg & Kumar, 2005*; *Galtier & Daubin, 2008*). In contrast to a previous similar study by *Thiergart, Landan & Martin (2014)*, comparison of the single gene trees were specifically performed with backbone nodes (excluding taxa closer to the tips of the trees) with the aid of the query hypotheses, which allowed us to interrogate each node and its associated signal manually. Although *Thiergart, Landan & Martin (2014)* also compared nodes between concatenated trees and the single gene phylogenies, their comparisons were focussed only on the recovery of identical nodes, which likely overestimated the effect of finer differences between the trees, leading them to their conclusion that the signal associated with the backbone or deeper nodes of their concatenated phylogenies are not preserved in single gene trees. Based on our data, three of the query hypotheses evaluated (see Fig. 3B and Fig. S1) had a large proportion of individual gene trees that did not support the expected monophyly of the taxon groups specified. These were hypothesis 5 in which *P. septica* is a singleton taxon placed as sister to the *P. agglomerans* and *P. ananatis* lineages (48.9% trees), hypothesis 9 in which *P. cypripedii* and *Pantoea* sp. At-9b are placed as sister to the *P. rodasii* and *P. dispersa* lineages (40.4% trees) and hypothesis 10 in which the *P. rodasii* and *P. dispersa* lineages are placed as sister groups (35.2% trees). In these instances, limited species have been sampled for the respective lineages. This undersampling of the diversity may contribute to the lack in robust recovery of the lineages due to large systematic error in the smaller datasets. In future, increased taxon sampling may resolve these problematic relationships more accurately in smaller datasets like those employed for the single gene trees (*Hedtke, Townsend & Hillis, 2006*; *Pollock et al., 2002*).

Comparison of single gene trees with phylogenies obtained from concatenated datasets, presents both a philosophical and logical quandary. It is widely accepted that single gene phylogenies, often with very limited or no statistical support, cannot be equated to a species phylogeny (*Degnan & Rosenberg, 2006*; *Doyle, 1992*; *Maddison, 1997*; *Pamilo & Nei, 1988*; *Rosenberg, 2002*). Despite the common practice of evaluating the robustness of a species phylogeny constructed from thousands or millions of characters, by its topological congruence to single gene trees (*Bapteste et al., 2009*; *Ciccarelli et al., 2006*; *Dagan & Martin, 2006*), these phylogenies are clearly not directly comparable and no conclusions regarding species evolution should be drawn from raw tree topology comparisons. This rationale is like comparing single molecules with chemical compounds and being disappointed that they do not share the same characteristics. Based on our data, the signal required for reconstructing a species phylogeny is dispersed and the only appropriate comparison of single gene trees to species trees would be when focus is placed on the evolution of a

particular gene or when species trees are inferred from single gene trees, as with the MSCM analyses.

Our findings confirm the robustness of phylogenies constructed from genomic data, based on the synergistic effect of combined genes, despite high levels of incongruence between individual gene trees. This is due to the phylogenetic signal for different nodes within the species phylogeny being distributed across the genome at higher levels than the randomly distributed conflicts within the dataset. These findings support previous conclusions suggested by several authors (*Andam & Gogarten, 2011*; *Daubin, Gouy & Perriere, 2002*; *Galtier & Daubin, 2008*; *Retchless & Lawrence, 2010*; *Rokas et al., 2003*), based on comparisons of single gene phylogenies with super trees and concatenated analyses using tree-to-tree distance approaches (*Daubin, Gouy & Perriere, 2002*; *Retchless & Lawrence, 2010*). Our results also suggest that the robustness of evolutionary hypotheses from whole genome data should be evaluated with phylogenetic network approaches that can depict conflicts, due to evolutionary processes or phylogenetic error, within the dataset (*Bryant & Moulton, 2002*; *Holland & Moulton, 2003*; *Huson & Bryant, 2005*). By employing such a total-evidence based approach, one would be able to recover a more realistic evolutionary hypothesis, particularly in terms of the deeper relationships, that also serves as an initial indication of the impact of organism-level evolutionary processes. Ultimately, such detailed evolutionary analyses would be invaluable for understanding the speciation process and for studying the development and distribution of important biological characteristics. Furthermore, our data also suggests that alternative approaches, focussing specifically on organism-level evolutionary processes, possibly at the population level, may be required to resolve relationships and elucidate the evolutionary history of younger taxa or leaves, where these processes may be rampant and phylogenetic incongruence highly prevalent.

## CONCLUSIONS

We found that phylogenetic conflict, potentially caused by organism-level evolutionary processes, was present in our phylogenomic dataset at both the gene partition and nucleotide levels. Although this non-phylogenetic signal could result from organism-level evolutionary process, like HGT and ILS, more in-depth analyses are needed to differentiate between these processes and to quantify the overall impact of these processes on the evolutionary history of the taxa. From our results it appeared that both signal and noise are randomly distributed across the genome and that all genes included in a concatenation-based phylogenomic analysis contribute signal toward the possible species tree. In other words, for *Pantoea* at least, phylogenies constructed from concatenated datasets are not driven by single genes, but rather that the signal from individual genes work synergistically to provide robust phylogenies.

## ACKNOWLEDGEMENTS

We would like to sincerely thank the reviewers, Dr. David Waite, Dr. Luis M. Rodriguez-R and an anonymous reviewer for valuable suggestions and significant contributions toward improving the manuscript.

### Funding

This work was supported by the National Research Foundation (South Africa), the DST-NRF Centre of Excellence in Tree Health Biotechnology (CTHB) and the University of Pretoria with regards to student funding. Informatics infrastructure was funded by the NRF National Bioinformatics Functional Genomics Grant (No: 93668). The funders had no role in study design, data collection and analysis, decision to publish, or preparation of the manuscript.

### Grant Disclosures

The following grant information was disclosed by the authors:
National Research Foundation (South Africa).
DST-NRF Centre of Excellence in Tree Health Biotechnology (CTHB).
University of Pretoria NRF National Bioinformatics Functional Genomics: 93668.

### Competing Interests

The authors declare there are no competing interests.

### Author Contributions

- Marike Palmer conceived and designed the experiments, performed the experiments, analyzed the data, contributed reagents/materials/analysis tools, prepared figures and/or tables, authored or reviewed drafts of the paper, approved the final draft.
- Stephanus N. Venter, Emma T. Steenkamp and Martin P.A. Coetzee conceived and designed the experiments, performed the experiments, analyzed the data, prepared figures and/or tables, authored or reviewed drafts of the paper, approved the final draft.
- Alistair R. McTaggart performed the experiments, analyzed the data, authored or reviewed drafts of the paper, approved the final draft.
- Stephanie Van Wyk, Juanita R. Avontuur, Chrizelle W. Beukes, Gerda Fourie, Quentin C. Santana and Magriet A. Van Der Nest performed the experiments, analyzed the data, approved the final draft.
- Jochen Blom performed the experiments, contributed reagents/materials/analysis tools, approved the final draft.

### Data Availability

A simple Python script was written to construct the individual phylogenies with FastTree v. 2.1 for the 1,357 protein sequences shared by all taxa analysed.

Two nexus files for the SplitsTree analyses are available as Supplemental Files.

## Supplemental Information

Supplemental information for this article can be found online at http://dx.doi.org/10.7717/peerj.6698#supplemental-information.

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
