# Peer review of "The synergistic effect of concatenation in phylogenomics: the case in Pantoea"

_PeerJ, doi:10.7717/peerj.6698_

## Round 0.1 · original submission · Major Revisions

All 3 reviewers are largely positive, but suggest a number of additional analyses. I agree that these analyses might significantly strengthen the manuscript and make it much more influential. On the other hand, I do not insist on performing all of the suggested analyses, as I understand that it might go far beyond the scope of the present study - I leave it to the authors to decide. However, in any case the missing alternatives need to be discussed in the text.

·

Basic reporting

The manuscript is very well written, and makes use of a well-characterised case study to address a topic of current relevance to microbial phylogenetics. The figures are informative and not excessive (i.e. only used to highlight relevant points in the manuscript) and there is generally ample citation to past research and literature. All results and discussion are clearly framed in the context of the research questions. Given that the manuscript focuses on a single genus, I feel that the title is too broad in scope (this would be more of a case study than a universal finding), but this is my only criticism of the writing.

The authors have also included (in my opinion) one of the best justifications for concatenated trees in the context of incomplete lineage sorting - that a speciation event does not mean that every gene in the organism diversifies, and hence the true evolutionary pattern of an organism may be distributed unevenly across many genes, which may not be captured in aggregate single-gene trees.

As a small note, Figure 3 is referred to before Figure 2 in the text of the manuscript, although it was probably intended that Figure 2 be first cited somewhere in LN215-220.

Experimental design

The experimental design and statistical analysis are rigorous and adequate to tackle the hypotheses proposed. The overall approach is also logically broken down into three main testable hypotheses, each of which is further subdivided as required. My only concern with the approach of the manuscript is that findings are evaluated in comparison to a Pantoea species hypothesis that is grounded in a concatenated tree. While I accept that we can never know the evolutionary history of a lineage it does seem circular to assume the concatenated model is true and to then evaluate the individual genes in the context of this model. The incorporation of independent lines of evidence to place weight on the concatenated model would add to the manuscript even if they are not specifically phylogenetic means. For example, genome-genome ANI or synteny comparisons could be used to predict the branching order of Pantoea species, against which both the concatenated and single-gene approaches could be compared.

This is particularly relevant in light of the work of Thiergart et al. (2014; https://www.ncbi.nlm.nih.gov/pmc/articles/PMC4302582). While it could certainly be argued that the authors are providing an explanation for why the 'disappearing tree' phenomenon occurs, it would be more powerful to control for this criticism of concatenated trees within the methods of the study.

Validity of the findings

The largest criticism of the results is likely that the authors have relied on approximate maximum likelihood trees (i.e. FastTree) as the foundation of their analysis. While they have provided a justification for this choice, and have verified findings using the more rigorous RAxML program where uncertainty arises, there are other methods of tree inference which can match or surpass the quality of RAxML with much lower compute times (for example, ExaML and IQ-Tree). FastTree also supports a very restricted range of evolutionary models (JTT, WAG, and LG) which may lead to reduced accuracy in the findings.

Reviewer 2 ·

Basic reporting

Some minor comments

* Line 103: I think that saying that the analysis accounted for all known phylogenetic errors is a bit of an overstatement. This sentence could be probably rewritten in a less

* Line 14-145 How much was "not robustly supported". Was there a bootstrap analysis? How much was the cutoff? --- PS (I see in the figure that this was a SH test, but how much was the value of the non-highly supported nodes?)

* Line 197-198 and the reference to Figure S1. This figure is too important to be left as the first figure in the Supplementary materials. It would stand to reason to make a larger figure 1 that includes in the first section the species tree and in the second section the net analysis so that both objects can be inspected visually right away. I would understand the decision of leaving the Species Tree for the SM only if the inclusion of it in Figure 1 makes the Figure to hard to read. Maybe some color guidelines below the main groups would be helpful

* Lines 211-212 The assertion would be easier to verify by having the two structures next to each other, see previous comment

* Lines 214-215 It is probably worth quoting here “Concatenated alignments and the case of the disappearing tree”. Thiergart et al. use a very similar approach you are using. It would be worth also pointing out the differences (which explain why yours make sense and theirs do not)

* Phylograms for limited sets of genes: I feel it is really a pity that the authors did not present their results in a better way. Please, see my comment in the section: Experimental design

* Please, share the file to open the SplitTree analysis

* Please, discuss (much more), the way that people are dealing today with phylogenetic incongruence (reconciliations). See the other sections

Experimental design

I have two major comments:

* Phylograms from limited set of genes

Although the approach that the authors used is not bad, I think that they could improve it quite a bit and obtain a potentially very robust and interesting result.

With the same experimental setup (20, 50 and 100 genes), obtaining the consensus phylogeny and then measuring the RF distance of the “true” species tree. Then plotting that to see the increasing trend in the similarity with the "true" species tree when more genes are sampled. Using more categories (e.g. 20, 40, 60, 80, 100 and 120) would probably allow the authors to obtain a very neat plot in which they can try to adjust a regression model. If the findings of the current paper are true, we shouldn’t see a great improvement by randomly sampling more than 100 gene trees.

* What about phylogenetic reconciliations?

One of the most popular ways of dealing with phylogenetic incongruence is using phylogenetic reconciliations. I was very disappointed to see that the authors completely left out this whole topic. For example, Szollosi et al 2012 (Phylogenetic modeling of lateral gene transfer reconstructs the pattern and relative timing of speciations) use a probabilistic reconciliation model that accounts for the phylogenetic incongruence to recover the species tree phylogeny. Even if the gene topologies are very often completely different from those of the species tree, the authors managed to recover a robust tree of cyanobacteria. I think a proper discussion on this topic is completely missing, as well as a proper analysis using some of these techniques to really quantify how many transfers can be found in this dataset (especially relevant when the authors set as one of the main goals of the paper, in line 113, to determine whether the observed conflicts could be ascribed to organism-level evolutionary processes). it is possible to answer this question using some of these techniques. See also Williams et al 2017 (Integrative modeling of gene and genome evolution roots the archaeal tree of life) used a phylogenomic analysis to study the evolution of Archaea genomes in which they dealt with the phylogenetic incongruence.

The authors might argue that they deal with HGT and ILS when using the network approach, but I would slightly disagree. They do indeed deal with this issue but in an insufficient manner. For example, the discussion in the lines 325-336 says that if more transfers or ILS were affecting this dataset, we would expect to see more boxes in the network analysis. I find this disappointing. I wonder if there is not a more numerical way of expressing these results instead of the highly subjective “low” or “high” number of boxes. Reconciliations analysis can give very precise estimates on the number of transfers affecting the dataset.

Validity of the findings

My biggest concern with this paper is that I feel that the importance of phylogenetic incongruence has not been dealt with properly. The network analysis does not show much more than saying than there is some incongruence between the gene trees and the species tree and that the major relationships are still preserved. This is not a surprising result.

The paper would benefit much also from a better analysis in the section Phylograms from limited set of genes

Additional comments

On the good side, I would point out that the authors deal with a very interesting topic and they performed a very exhaustive work in some areas. They explored the impact of recombination in their datasets, the origin of the phylogenetic incongruence at various level and the possiblity of obtaining a good species tree even when there is no apparent signal in the individual gene trees.

On the "down" side, I feel that the HGT issue was completely left aside, that they did not discuss a major topic on the uses of phylogenetic incongruence and that the approach of some of the experiments they carried out should have been presented in a more numeric way.

·

Basic reporting

No comment.

Experimental design

No comment.

Validity of the findings

In general, the work in this manuscript is well described, the aims clear, and the approaches sound. However, I would like to bring an issues to the authors' attention: In my opinion the authors overstate the accuracy of Maximum Likelihood (or AML) methods for phylogenetic reconstruction based on concatenation. It is true that ML/concatenation is often regarded as a nearly infalible gold standard in phylogenomic studies where the final tree is the main objective. However, this manuscript being concerned with the statistical robustness of the methods applied (rather than the final tree), should at least consider the observation that such methods are statistically inconsistent. This is problematic in the current manuscript in at least two points:

(1) The authors appear to take the concatenation-based phylogenetic reconstruction in Palmer et al (2017) as incontrovertible truth. Although I consider that work to present a high-quality phylogeny of the group, it remains a (likely) phylogenetic hypothesis. I'd like to invite the authors to consider the possibility that their previously published reconstruction be flawed.

(2) The authors do not consider other methods that could account for ILS, such as those based on multi-species coalescent model. It would be fantastic to see these methods used side to side in this study. However, if that's simply beyond the scope of the current study, the scope of the study as stated must be revised and this limitation should be made explicit. For example, note that the authors generate reconstructions with different numbers of loci and compare those to the reconstruction generated with all core genes of the larger group in Palmer et al (2017; 1,039 genes). This comparison is valuable, as it tells at which point the concatenation-based reconstruction saturates (i.e., it doesn't change anymore with additional loci), but it cannot be taken as a metric of loci necessary to reconstruct the true species tree (i.e., it's error-free). There is good evidence in the literature that ML/concatenation doesn't necessarily tend towards the true species tree regardless of the number of loci; i.e., it's statistically inconsistent. This is not true for MSCM, which is guaranteed to tend towards the species tree when the number of loci increases given true gene trees (although the absolute error may at some points be higher than that of ML/concatenation).

Additional comments

Minor issues detected:

L115: contains -> contain
L136: extra comma
L144: "When the relationships obtained from AML analyses were not robustly supported, ...": please make the criterion used explicit. Was this based on the bootstrap supports? Did the authors use a minimum, maximum, or average node support as the threshold?
L165: was -> were
L199 and elsewhere: Alignment lengths should be given in "columns" or "amino acid columns", unless the authors mean to indicate all characters in the alignment matrix excluding gaps and missing data
L329-334: This section appears very subjective. Perhaps the authors could provide more quantitative statements to strengthen this section
L384-386: "... [gene and species] phylogenies are clearly not comparable and no conclusions regarding species evolution should be drawn from such comparison": I suspect the authors mean that direct comparison (e.g., raw tree topology distances) shouldn't be considered as robust tests for the quality of species tree reconstructions. However, I must disagree with the statement as is. Gene and species phylogenies are comparable: indeed, that's the fundamental idea behind MSCM-based reconstruction, to compare gene trees to (potential) species trees in order to reduce deep coalescence.

Signed: Luis M. Rodriguez-R

---

## Round 0.2 · Minor Revisions

There are minor editorial comments that need to be addressed prior to acceptance.

·

Basic reporting

No comment

Experimental design

No comment

Validity of the findings

No comment

Additional comments

The authors have addressed my previous concerns. I have no further comments or criticisms of the manuscript.

Reviewer 2 ·

Basic reporting

I feel that the paper has improved much after the reviews. I especially enjoyed the discussion of Thiergart et al. 2014. I want to congratulate the authors for the good job

Experimental design

No comment

Validity of the findings

No comment

Additional comments

No comment

·

Basic reporting

I would like to echo Dr. Waite's observation on the title. Although this study may serve as a specific example that can be used more broadly, all of it is circumscribed to the genus Pantoea. Not only are all the results derived from a single genus, but nothing in the discussion attempts to address potential differences in other taxa, making this manuscript very specific to Pantoea. I respectfully invite the authors to reconsider the title. I have no other comments on the basic reporting.

Experimental design

No comment.

Validity of the findings

No comments.

Additional comments

The authors repeatedly indicate that the branch lengths in the MSC tree indicate the discordance between gene trees. I don't believe this is accurate, and is not supported by the cited reference. The coalescent units are simply mutation units scaled by population sizes. Since population sizes are not necessarily constant, the relationship is not necessarily linear, but they're not a measurement of discordance between gene trees.

On a related note, the new Figure 1 presents the MSC tree. In the tree the authors decided to present terminal branches with a length of 1 coalescent unit. This decision should be indicated in the legend. Note that ASTRAL does not (cannot) compute lengths of terminal branches.

Signed: Luis M. Rodriguez-R

---

## Round 0.3 · accepted · Accept

All remaining concerns have been addressed.

#